# SLC35G1 is a highly chloride-sensitive transporter responsible for the basolateral membrane transport in intestinal citrate absorption

**Yoshihisa Mimura[1][†], Tomoya Yasujima[1]\*, Katsuhisa Inoue[2], Shogo Akino[1], Chitaka Namba[1], Hiroyuki Kusuhara[3], Yutaro Sekiguchi[1], Kinya Ohta[4], Takahiro Yamashiro[1], Hiroaki Yuasa[1]**

[1]Department of Biopharmaceutics, Graduate School of Pharmaceutical Sciences, Nagoya City University, Nagoya, Japan; [2]Department of Biopharmaceutics, School of Pharmacy, Tokyo University of Pharmacy and Life Sciences, Tokyo, Japan; [3]Laboratory of Molecular Pharmacokinetics, Graduate School of Pharmaceutical Sciences, The University of Tokyo, Tokyo, Japan; [4]College of Pharmacy, Kinjo Gakuin University, Nagoya, Japan

**\*For correspondence:**
yasujima@phar.nagoya-cu.ac.jp

**Present address:** [†]Department of Pharmacy, Nagoya City University Hospital, Nagoya, Japan

**Competing interest:** The authors declare that no competing interests exist.

## eLife Assessment

This work identifies the molecular function of an orphan human transporter, SLC35G1, providing **convincing** evidence that this protein is involved in intestinal citrate absorption. This work provides **important** insight into transporter function and human physiology.

**Abstract** The intestinal absorption of essential nutrients, especially those not readily biosynthesized, is a critical physiological process for maintaining homeostasis. Numerous studies have indicated that intestinal absorption is mediated by various membrane transporters. Citrate, a crucial bioactive compound produced as an intermediate in the Krebs cycle, is absorbed in the small intestine through carrier-mediated systems because of its high hydrophilicity. While the luminal absorption of citrate is mediated by $Na^+$-dicarboxylate cotransporter 1 (NaDC1/SLC13A2), the mechanism governing the release of the transported citrate into the bloodstream remains unknown. Here, we explored the transporters responsible for intestinal citrate absorption at the basolateral membrane, focusing on highly expressed orphan transporters in the small intestine as candidates. Consequently, SLC35G1, originally identified as a partner of stromal interaction molecule 1, a cell surface transmembrane glycoprotein, was found to play a role in the intestinal absorption of citrate at the basolateral membrane. Furthermore, our results revealed that SLC35G1-mediated citrate transport was diminished by chloride ions at physiologically relevant extracellular concentrations. This suggests that SLC35G1, to our best knowledge, is the first transporter identified to be extremely sensitive to chloride ions among those functioning on the basolateral membrane of intestinal epithelial cells. This study provides valuable insights into the intestinal absorption of citrate and significantly contributes to elucidating the poorly understood molecular basis of the intestinal absorption system.

## Introduction

Citrate plays a crucial role as a tricarboxylic acid (TCA) cycle intermediate, contributing to the generation of metabolizable energy. Since citrate is obtained not only through biosynthesis but also from the diet, the existence of an intestinal absorption system has long been acknowledged (*Browne et al., 1978*). Luminal absorption is predominantly mediated by Na⁺-dependent dicarboxylate transporter 1 (NaDC1/SLC13A2) *Pajor, 2006*; however, the transporter responsible for the release of citrate and its metabolites across the basolateral membrane remains unidentified (*Pajor, 2014*). SLC62A1, which was initially recognized as a pyrophosphate transporter linked to joint mineralization, has recently been reported to facilitate citrate efflux in some cell types (*Szeri et al., 2020*). However, it is important to note that the role of SLC62A1 has been mainly implicated in osteoblasts, prostate, skeletal muscle, brain, and the cardiovascular system. It may have the major role in the systemic disposition of citrate rather than the intestinal absorption. To elucidate the molecular mechanism underlying citrate absorption at the basolateral membrane, we explored the transporters implicated in the process. Our study provides novel insights into basolateral membrane transporters, addressing a previously poorly understood aspect in intestinal epithelial cells.

## Results and discussion

To identify a novel citrate transporter expressed in the small intestine, we conducted a comprehensive in silico screening to narrow down candidate orphan transporters. This initial screening involved bioinformatics analysis of gene expression database to identify candidate transporters that showed significant expression in intestine but were not or little characterized previously. Following the in silico screening, we cloned the candidate transporters, had them transiently expressed in human embryonic kidney 293 (HEK293) cells, and evaluated them for citrate transport activity by [¹⁴C]citrate uptake assay. Through this process, we discovered that SLC35G1, a member of the solute carrier (SLC) 35 family and originally identified as a partner of stromal interaction molecule 1 (*Krapivinsky et al., 2011*), is capable of transporting citrate at the plasma membrane. While the functions of nucleoside-sugar transporters on the endoplasmic reticulum membrane have been identified for certain SLC35 family members, including SLC35A, SLC35B, SLC35C, and SLC35D (*Ishida and Kawakita, 2004*), the roles of SLC35E, SLC35F, and SLC35G have not yet been fully elucidated, as shown in *Figure 1A*. SLC35G1 was indicated to be mainly localized to the endoplasmic reticulum (ER) in the earlier study, in which SLC35G1 was tagged with GFP. A possibility is that SLC35G1 was wrongly directed to ER due to the modulation in the study. For the functional analysis, we first established Madin-Darby canine kidney (MDCKII) cells stably expressing SLC35G1. Using these cells, we observed a linear increase in SLC35G1-mediated citrate uptake with time, specifically during the first 10 min in Hanks' solution mimicking the cytosolic ion content by eliminating Cl⁻ (*Figure 1B*). The acidic condition (pH 5.5) was used to gain sufficient uptake for functional analysis. Therefore, the uptake at 10 min was evaluated in subsequent studies. A kinetic analysis revealed that the SLC35G1-specific uptake of citrate exhibited saturation kinetics, with a $V_{max}$ of 1.10 nmol/min/mg protein and a $K_m$ of 519 µM (*Figure 1C*). Notably, the specific uptake of citrate by SLC35G1 was extensively inhibited by extracellular Cl⁻ (*Figure 1D*), with an $IC_{50}$ value of 6.7 mM (*Figure 1E*). As previously reported (*Melkikh and Sutormina, 2008*), the Cl⁻ concentration is lower in the cytosol (5 mM) than in the extracellular conditions (120 mM). Therefore, SLC35G1 could be a potential citrate exporter. It is notable that, as shown in *Figure 1D*, the use of K-gluconate in place of Na-gluconate, which induces plasma membrane depolarization, had no impact on the specific uptake, suggesting that SLC35G1-mediated citrate transport is independent of membrane potential. Additionally, we examined the effect of various compounds (200 µM), to explore the potential role of SLC35G1 in their transport (*Figure 1F*). The tested compounds were TCA cycle intermediates (fumarate, α-ketoglutarate, malate, pyruvate, and succinate) as substrate candidate carboxylates analogous to citrate, diverse anionic compounds (BSP, DIDS, probenecid, pravastatin, and taurocholate) as those that might be substrates or inhibitors, and diverse cationic compounds (cimetidine, quinidine, and verapamil) as those that are least likely to interact with SLC35G1. Among them, certain anionic compounds significantly reduced SLC35G1-specific citrate uptake, suggesting that they may interact with SLC35G1. However, we could not identify any structural features commonly shared by these compounds, except that they have anionic moieties. At the concentration of 200 µM, competing substrates with an affinity comparable to that of citrate could inhibit SLC35G1-specific

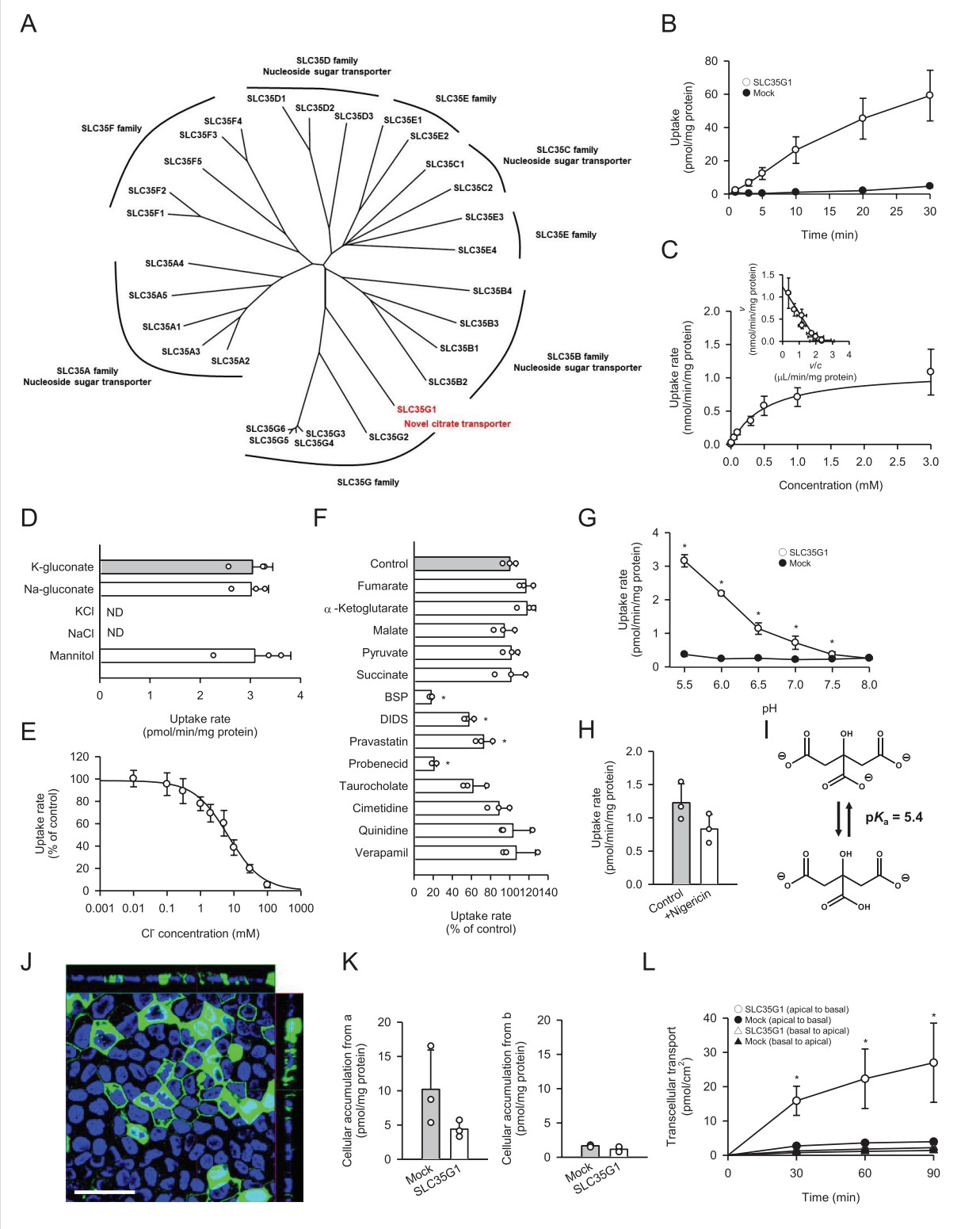

**Figure 1.** Functional characteristics of SLC35G1 stably expressed in MDCKII cells. (**A**) Phylogenetic tree of the SLC35 members and identified functions. (**B**) Time course of [$^{14}$C]citrate (1 μM) uptake in SLC35G1-transfected MDCKII cells (open circles) and mock cells (closed circles), evaluated at pH 5.5 and 37°C under the Cl$^-$-free condition. (**C**) Concentration-dependent [$^{14}$C]citrate uptake by SLC35G1, evaluated at various concentrations for 10 min at pH 5.5 and 37°C under the Cl$^-$-free condition. The estimated values of $V_{max}$ and $K_m$ were 1.11±0.16 nmol/min/mg protein and 519±86 μM, respectively. (**D**) Effect of ionic conditions on [$^{14}$C]citrate (1 μM) uptake by SLC35G1, evaluated for 10 min at pH 5.5 and 37°C. K-gluconate in the control uptake

*Figure 1 continued on next page*

*Figure 1 continued*

solution was replaced as indicated. (**E**) Concentration-dependent chloride inhibition of [$^{14}$C]citrate (1 µM) uptake by SLC35G1, evaluated at various chloride concentrations for 10 min at pH 5.5 and 37°C. The estimated value of $IC_{50}$ was 6.7±1.4 mM. (**F**) Effect of various compounds on [$^{14}$C]citrate (1 µM) uptake by SLC35G1, evaluated for 10 min at pH 5.5 and 37°C in the presence (200 µM) or absence (control) of a test compound under the Cl$^-$-free condition. BSP, bromosulfophthalein; DIDS, 4,4'-diisothiocyano-2,2'-stilbenedisulfonic acid. (**G**) Effect of pH on [$^{14}$C]citrate (1 µM) uptake in SLC35G1-transfected MDCKII cells (open circles) and mock cells (closed circles), evaluated for 10 min at 37 °C under the Cl$^-$-free condition, using the uptake solutions supplemented with 10 mM MES (pH 6.5 and below) or 10 mM HEPES (pH 7.0 and above). (**H**) Effect of the protonophore nigericin on [$^{14}$C] citrate (1 µM) uptake by SLC35G1, evaluated for 10 min at pH 5.5 and 37°C under the Cl$^-$-free condition in the presence (10 µM) or absence (control) of nigericin after pretreatment for 5 min with or without nigericin under the same conditions. (**I**) Structural formula illustrating the chemical equilibrium of citrate. (**J**) Cellular localization of SLC35G1 stably expressed in polarized MDCKII cells. The confocal laser-scanning microscopic image shows SLC35G1 (green) and nuclei stained with 4',6-diamidino-2-phenylindole (DAPI; blue). Scale bar, 50 µm. (**K**) Intracellular accumulation of citrate in polarized MDCKII cells stably expressing SLC35G1 and mock cells on Transwell filters. Intracellular accumulation of [$^{14}$C]citrate (1 µM) was evaluated for 90 min at pH 5.5 on the apical side (**a**) and pH 7.4 on the basolateral side (**b**) in the presence of Cl$^-$ (144 mM) on both sides of intestinal epithelial cells. (**L**) Time course of transcellular [$^{14}$C]citrate (1 µM) transport in polarized MDCKII cells stably expressing SLC35G1 and mock cells on Transwell filters, evaluated at pH 5.5 in the apical chamber and pH 7.4 in the basolateral chamber in the presence of Cl$^-$ (144 mM) in both chambers. Data represent the mean ± SD of three biological replicates (**B–H, K, L**). Statistical differences were assessed using analysis of variance (ANOVA) followed by Dunnett's test (**D, F, L**) or using Student's t-test (**G, H, K**). *$p<0.05$ compared with the control (**F**), or compared with the mock at each pH (**G**) or time point (**L**).

citrate uptake by about 30%. Therefore, it is likely that the compounds that did not exhibit a significant effect have no affinity or at least a lower affinity than citrate to SLC35G1. SLC35G1-specific citrate uptake showed pH dependence (*Figure 1G*). To investigate whether SLC35G1 utilizes an H$^+$ gradient as a driving force for citrate transport, we examined the effect of the protonophore nigericin in the uptake solution at pH 5.5 to dissipate the H$^+$ gradient across the plasma membrane. However, treatment with nigericin did not affect the specific uptake of citrate (*Figure 1H*), excluding this possibility. Based on these, it is likely that SLC35G1 is a facilitative transporter, although we cannot exclude another possibility that its sensitivity to extracellular Cl$^-$ might imply its operation as a citrate/Cl$^-$ exchanger. Given that one carboxylate can be protonated with a p$K_a$ of 5.4 (*Contreras-Trigo et al., 2018*; *Figure 1I*), SLC35G1 may preferably transport dianionic citrate, like NaDC1 (*Pajor and Sun, 1996*). Finally, we determined the role of SLC35G1 in the transcellular transport of citrate at the basolateral membranes using SLC35G1-transfected polarized MDCKII cells on Transwell permeable membrane filters. As shown in *Figure 1J*, SLC35G1 predominantly localized to the basolateral membrane of polarized MDCKII cells. Transport studies were performed under pH gradient conditions (apical pH 5.5 and basal pH 7.4) in the presence of Cl$^-$ (144 mM) to mimic intestinal physiological conditions. SLC35G1-transfected cells showed no significant change in intracellular citrate accumulation from the apical side, compared to mock cells (*Figure 1K*). However, apical-to-basolateral transcellular transport was significantly higher in SLC35G1-transfected cells (*Figure 1L*). On the other hand, SLC35G1 overexpression affected neither intracellular accumulation nor transcellular transport from the basolateral side. This is reasonable since the basolaterally localized SLC35G1 operates for efflux but not for uptake in the presence of Cl$^-$ in the extracellular environment. These results suggest that SLC35G1 plays a role in the intestinal absorption of citrate at the basolateral membrane as an exporter under physiological conditions.

To further investigate the physiological role of SLC35G1, we examined its expression in human tissues (*Figure 2A*). Quantitative real-time PCR analysis revealed that SLC35G1 is highly expressed in the digestive tract, especially in the upper part of the small intestine, including the duodenum and jejunum, followed by the testis and pancreas. Additionally, immunohistochemical staining of the human jejunum clearly showed that SLC35G1 is localized on the basolateral membrane, as indicated by the co-localization of ATP1A1, a marker protein on the basolateral membrane (*Figure 2B*). To investigate the physiological impact of SLC35G1 on intestinal citrate absorption, we examined the effect of SLC35G1 knockdown on citrate uptake in human colorectal adenocarcinoma Caco-2 cells, a commonly used model for studying intestinal absorbency (*Yee, 1997*; *Figure 2C*), in the absence of Cl$^-$. As shown in *Figure 2C*, citrate uptake was significantly reduced following RNA interference with siRNAs targeting SLC35G1 mRNA, providing evidence of the involvement of SLC35G1 in intestinal citrate absorption. The reduction in citrate uptake was not indicative of off-target effects, as multiple siRNAs with distinct nucleic acid sequences yielded similar reductions. Moreover, the expression of SLC35G1 mRNA was reduced by these siRNAs, confirming successful silencing of SLC35G1 (*Figure 2D*). The correlation between the decrease in citrate uptake and mRNA levels suggests that

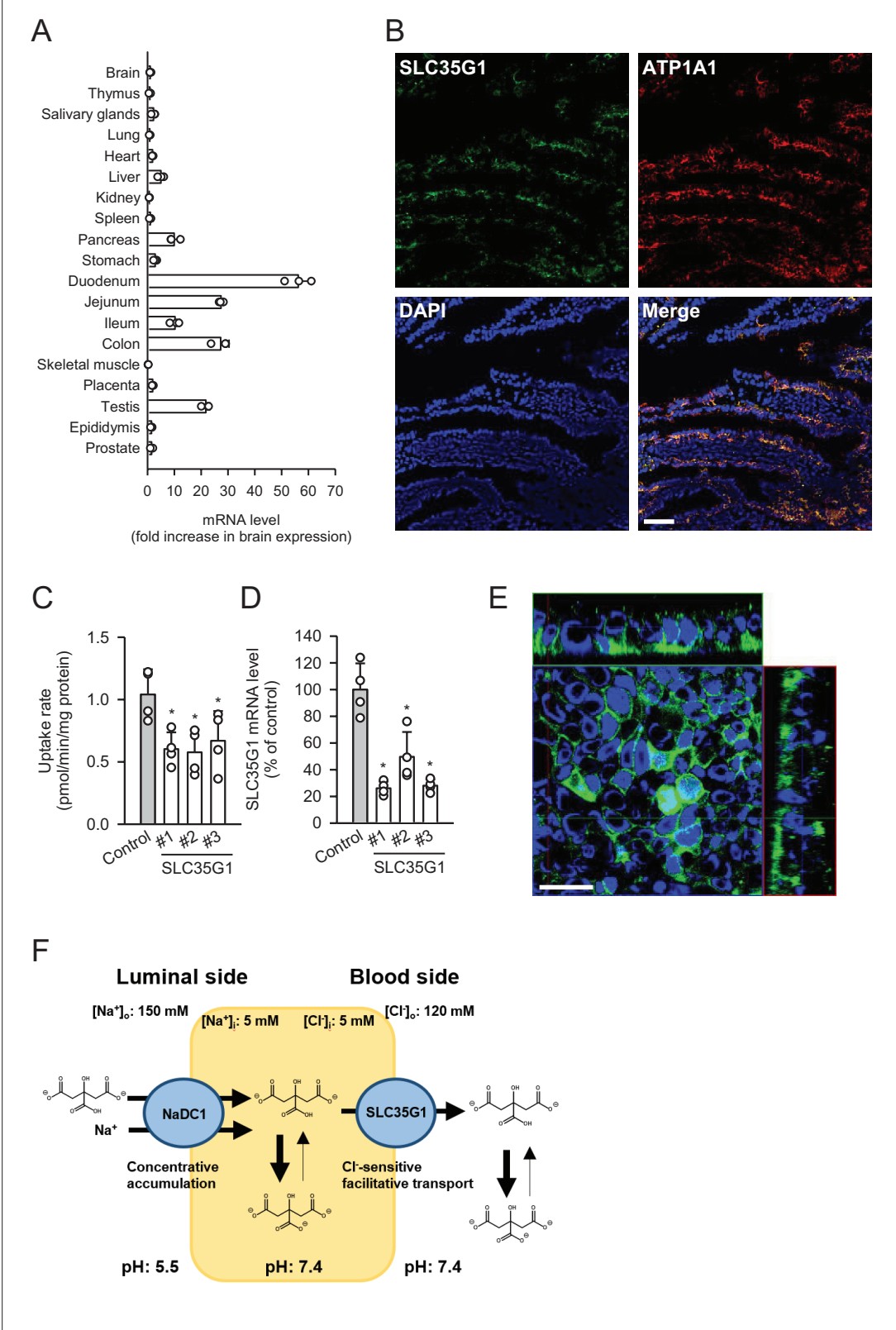

**Figure 2.** Role of SLC35G1 in intestinal citrate absorption. (**A**) Analysis of SLC35G1 mRNA expression in various human tissues using quantitative real-time PCR. (**B**) The immunofluorescent images show the subcellular localization of SLC35G1 (green), ATP1A1 (red), and 4',6-diamidino-2-phenylindole (DAPI; blue) in the human jejunum. Scale bar, 100 μm. (**C–D**) Effect of SLC35G1 silencing on citrate uptake and on SLC35G1 mRNA expression in Caco-2 cells. (**C**) The uptake rate of [$^{14}$C]citrate (1 μM) was evaluated for 10 min at pH 5.5 and 37°C under the Cl$^-$-free condition in Caco-2 cells transfected with

*Figure 2 continued on next page*

*Figure 2 continued*

three different siRNAs specific to SLC35G1 mRNA or control siRNA. (**D**) SLC35G1 mRNA levels were assessed using quantitative real-time PCR analysis. (**E**) Cellular localization of endogenous SLC35G1 in polarized Caco-2 cells. The confocal laser-scanning microscopic image shows SLC35G1 (green) and nuclei stained with DAPI (blue). Scale bar, 50 µm. (**F**) Schematic model showing intestinal citrate absorption mediated by NaDC1 and SLC35G1. Data represent the mean ± SD of three technical repeats (**A**) or biological repeats (**C–D**). Statistical differences were assessed using analysis of variance (ANOVA) followed by Dunnett's test (**C–D**).

SLC35G1 is primarily involved in citrate uptake. Immunohistochemical staining revealed that endogenous SLC35G1 in polarized Caco-2 cells cultured on Transwell membranes was also localized on the basolateral membranes (*Figure 2E*). Taken together, these results indicate that intestinal citrate transport at the basolateral membrane is primarily mediated by SLC35G1. Additionally, it acts cooperatively with NaDC1, which is reported to be highly expressed in the jejunum (*Pajor, 1995*), for citrate absorption on the apical membrane, as illustrated in *Figure 2F*.

Our findings suggest that SLC35G1 is localized on the basolateral side of intestinal epithelial cells and, to our knowledge, is the first transporter identified with chloride sensitivity, which is a unique characteristic to render with the directional transport function for an SLC transporter. These findings pave the way for several follow-up directions related to the molecular basis of the intestinal absorption of various compounds, including nutrients and drugs.

## Materials and methods

### Materials

[$^{14}$C]citrate (116.4 mCi/mmol) was obtained from PerkinElmer Life and Analytical Sciences (Boston, MA, USA). Unlabeled citrate and Dulbecco's modified Eagle's medium (DMEM) were sourced from Wako Pure Chemical Industries (Osaka, Japan), and fetal bovine serum (FBS) was obtained from Sigma Aldrich (St. Louis, MO, USA). All other reagents were of analytical grade and commercially acquired.

### Cell culture

Madin-Darby canine kidney II (MDCKII) cells and Caco-2 cells were obtained from RIKEN BioResource Research Center (Tsukuba, Japan) with catalog numbers RCB5148 and RCB0988, respectively. Both cells were cultured at 37°C and 5% $CO_2$ in DMEM supplemented with 10% FBS, 100 U/mL penicillin, and 100 µg/mL streptomycin, as previously described (*Mimura et al., 2017*), and were confirmed to be free of mycoplasma contamination.

### Preparation of plasmids

The cDNA for human SLC35G1 was cloned using a reverse transcription (RT)-PCR-based method, as previously described (*Mimura et al., 2017*). Briefly, an RT reaction was performed to obtain a cDNA mixture, using 1 µg of the human intestine total RNA (BioChain Institute, Newark, CA), an oligo(dT) primer, and the highly efficient reverse transcriptase ReverTra Ace (Toyobo, Osaka, Japan). The SLC35G1 cDNA was then amplified by PCR, using KOD plus polymerase (Toyobo) and the following primers: forward primer, 5'-GAGATGCGGCCTCAGGACAG-3', and reverse primer, 5'-TAGCCTCCCCACCCATATCCC-3'. These primers were designed based on the GenBank sequences for SCL35G1 (accession number: NM_001134658.2). The second PCR was performed using the initial PCR product as a template, with a forward primer containing an EcoRI restriction site (underlined), 5'-A<u>GAATTC</u>CGAGATGCGGCCTCAGGAC-3', and a reverse primer containing an XbaI restriction site (underlined), 5'-GTT<u>TCTAGA</u>TGTGTGCGTATGCTG-3'. The resulting cDNA product was inserted into the pCI-neo vector (Promega, Madison, WI, USA) between the EcoRI and XbaI sites, and the sequence of the final product was determined using an automated sequencer.

### Uptake study

First, we established MCDKII cells stably expressing SLC35G1 as previously described (*Yamamoto et al., 2010*). Briefly, MDCKII cells were transfected with the plasmid carrying the SLC35G1 cDNA, using Lipofectamine 2000 as a transfection reagent. The cells were then cultured in DMEM supplemented with 10% FBS and 800 µg/mL G418 for 2–3 weeks. Subsequently, G418-resistant clones were

selected and assessed for the transport of [$^{14}$C]citrate. MDCKII cells stably expressing SLC35G1 with high citrate transport activity were used in subsequent studies.

MDCKII cells stably expressing SLC35G1 ($1.5\times10^5$ cells/mL, 1 mL/well) were grown in 24-well plates for 48 h until they reached confluence. For standard transport assays, cells were preincubated in substrate-free uptake buffer, which was Hanks' solution modified to reflect cytosolic conditions (142.06 mmol/L K-gluconate, 0.812 mmol/L MgSO$_4$, 0.379 mM K$_2$HPO$_4$, 0.441 mM KH$_2$PO$_4$, 0.952 mM Ca-gluconate, 25 mM D-glucose) and supplemented with 10 mM 2-(N-morpholino)-ethanesulfonic acid (MES; pH 5.5), for 5 min. Subsequently, uptake assays were initiated by replacing the substrate-free uptake buffer with an uptake buffer containing a [$^{14}$C]citrate (0.25 mL). For initial uptake analysis experiments, the uptake period was set to 10 min in the initial phase, in which uptake was in proportion to time. Other experiments were conducted in a time-dependent manner, as specified. When examining the effects of various compounds on citrate uptake, the test compounds were added to the buffer only during the uptake period. All procedures were performed at 37°C. Assays were stopped by adding 2 mL of ice-cold substrate-free uptake buffer, followed by two washes with an additional 2 mL of the same buffer. The cells were solubilized in 0.5 mL of 0.2 mol/L NaOH solution containing 0.5% sodium dodecyl sulfate (SDS), and the associated radioactivity was measured by liquid scintillation counting for uptake evaluation. The cellular protein content was determined by the bicinchoninic acid (BCA) method, using bovine serum albumin as the standard (3). In experiments to examine the effect of ionic conditions, K-gluconate in the control uptake solution was replaced as indicated. Uptake solutions were supplemented with 10 mM MES (pH 6.5 and below) or 10 mM HEPES (pH 7.0 and above) in experiments to examine the effect of pH. In experiments to examine the effect of nigericin as an agent for intracellular acidification, a mannitol-based buffer (250 mM mannitol, 1.2 mM MgSO$_4$, 2 mM KH$_2$PO$_4$, 5 mM D-glucose, 10 µM nigericin, and 20 mM MES) was used and uptake assays were conducted for 10 min at pH 5.5 and 37°C in the presence (10 µM) or absence (control) of nigericin after pretreatment for 5 min with or without nigericin under the same conditions. To estimate nonspecific uptake, uptake assays were conducted in mock cells transfected with an empty pCI-neo vector. The specific uptake of citrate by SLC35G1 was estimated by subtracting the uptake in mock cells from that in SLC35G1-transfected cells.

## Transcellular transport study

MDCKII cells were seeded at a density of $2\times10^5$ cells on each polycarbonate membrane insert in a 12-well Transwell plate and cultured for 5 days. After removing the culture medium from both sides of the inserts, cells were preincubated for 5 min at 37 °C in Hanks' solution (136.7 mM NaCl, 5.36 mM KCl, 0.952 mM CaCl$_2$, 0.812 mM MgSO$_4$, 0.441 mM KH$_2$PO$_4$, 0.385 mM Na$_2$HPO$_4$, and 25 mM D-glucose) supplemented with 10 mM MES (pH 5.5) in the apical chamber and 10 mM HEPES (pH 7.4) in the basolateral chamber. To initiate transcellular transport from the apical side to the basolateral side, 0.5 mL of the Hanks' solution (pH 5.5) containing [$^{14}$C]citrate was replaced in the apical chamber. For transport in the opposite direction (basolateral side to apical side), 1.5 mL of Hanks' solution (pH 7.4) containing [$^{14}$C]citrate was replaced in the basal chamber. To monitor transcellular transport, samples (100 µL from basal chamber and 50 µL from the apical chamber) were periodically collected, replenishing each collected volume with an equal volume of fresh buffer. At the end of the transport study, the assays were stopped by adding ice-cold Hanks' solution (1 mL for the apical chamber and 3 mL for the basolateral chamber), followed by washing the cells twice with the same buffer. The cells were then solubilized in 0.5 mL of 0.2 M NaOH solution containing 0.5% SDS at room temperature for 1 hr. The radioactivity associated with the cells was measured by liquid scintillation counting for uptake evaluation.

## Quantification of SLC35G1 mRNA by quantitative real-time PCR

Total RNA samples from various human tissues (BioChain Institute) and Caco-2 cells were used to prepare cDNA using the ReverTra Ace reverse transcriptase. A quantitative real-time PCR was performed using SsoFast EvaGreen Supermix with Low ROX (Bio-Rad Laboratories, Hercules, CA, USA) on a 7300 Fast Real-time PCR System (Applied Biosystems, Foster City, CA, USA) with the following primers: forward primer for SLC35G1, 5'- AGAGCCCACTGAGAAAAGGA –3'; reverse primer for SLC35G1, 5'-GTAGGTGCTGGCTCTGCCT-3'; forward primer for GAPDH, 5'-CGGAGTCAACGG

ATTTGGTCGTAT-3′; reverse primer for GAPDH, 5′-AGCCTTCTCCATGGTGGTGAAGAC-3′. GAPDH was employed as an internal control for normalization of the mRNA expression levels of SLC35G1.

## Knockdown study

Caco-2 cells (5.0×10⁴ cells/mL, 1 mL/well) were grown on 24-well-coated plates for 6 hr, transfected with 5 pmol/well of the Silencer Selected siRNAs specific to the mRNA of SLC35G1 (Thermo Fisher Scientific, Waltham, MA), using 1.5 mL/well of Lipofectamine RNAi MAX (Thermo Fisher Scientific), and cultured for 5 days for silencing of the designated transporter by RNAi. The sequences of the siRNAs are following: sense sequence of #1 siRNA, GGAGUGAUCCUUAUCGUGATT; antisense sequence of #1 siRNA, UCACGAUAAGGAUCACUCCAG, sense sequence of #2 siRNA, CUUGCUUAAUAUACAG AAATT; antisense sequence of #2 siRNA, UUUCUGUAUAUUAAGCAAGGG, sense sequence of #3 siRNA, CAUUUGGUAUUAUGUAGUATT; antisense sequence of #3 siRNA, UACUACAUAAUACCAA AUGCT. For control, negative control Silencer Selected siRNA (Thermo Fisher Scientific), was used. Citrate uptake assays were conducted as described above for SLC35G1-transfected MDCKII cells.

## Immunofluorescence staining

To examine the localization of SLC35G1, MDCKII cells stably expressing SLC35G1 were seeded at a density of 2×10⁵ cells/insert on a 12-well Transwell plate and cultured for 5 days, and Caco-2 cells were seeded at a density of 5×10⁵ cells/insert on a 12-well Transwell plate and cultured for 21 days. The cells were washed thrice with ice-cold PBS and fixed in 5% paraformaldehyde for 15 min. After washing thrice with PBS, the cells were incubated for 1 h at room temperature with an anti-human SLC35G1 goat polyclonal antibody at a dilution of 1:100 in Can Get Signal Immunostain Immuno-reaction Enhancer Solution A (Toyobo). The cells were washed with PBS, and the primary antibody was probed with anti-goat IgG Alexa Fluor 488 (Jackson ImmunoResearch Laboratories, West Grove, PA, USA) at a dilution of 1:400 in Can Get Signal Immunostain Immunoreaction Enhancer Solution B (Toyobo) for 1 hr at room temperature. After three washes with PBS, the cells were mounted on a glass slide in 9:1 glycerol/PBS containing 1 µM DAPI.

A specimen of normal human adult small intestine frozen tissue sections (US Biomax, Derwood, MD, USA) was blocked in G-Block (GenoStaff, Tokyo, Japan) for 1 h and then incubated for 1 hr at room temperature with an anti-human SLC35G1 goat polyclonal antibody at a dilution of 1:100 in Can Get Signal Immunostain Immunoreaction Enhancer Solution A. After washing thrice with PBS, the primary antibody was probed with anti-goat IgG Alexa Fluor 488 by incubating for 1 hr at room tempera-ture. Additionally, we stained ATP1A1 with an anti-human ATP1A1 mouse monoclonal antibody (clone BXP-21, Abcam, Cambridge, UK) labeled with the Ab-10 Rapid Hilyte Fluor 647 labeling kit (Dojindo Laboratories, Kumamoto, Japan) according to the manufacturer's instructions. After washing thrice with PBS, the tissues were mounted in 9:1 glycerol/PBS containing 1 µM DAPI.

The localization of immunofluorescently labeled proteins and nuclei was visualized using a confocal laser scanning microscope (LSM510; Zeiss, Jena, Germany).

## Data analysis

The saturable transport of citrate by SLC35G1 was analyzed by assuming Michaelis-Menten-type carrier-mediated transport, as represented by the following equation: $v = V_{max} \times s/(K_m + s)$. The apparent parameters of the maximum transport rate ($V_{max}$) and Michaelis constant ($K_m$) were estimated by fitting this equation to the experimental profile of the uptake rate ($v$) versus the concentration ($s$) of citrate as the substrate, using a nonlinear least-squares regression analysis program, WinNonlin (Pharsight, Mountain View, CA, USA), with $v^{-2}$ as the weight.

When $s$ is considerably smaller than $K_m$ ($s << K_m$), $v$ in the presence of an inhibitor can be expressed as $v = v_0/(1 + (i/IC_{50})^n)$. The half-maximal inhibitory concentration ($IC_{50}$) was estimated, along with the Hill coefficient ($n$) and $v$ in the absence of inhibitors ($v_0$), by fitting this equation to the experimental profile of $v$ versus the inhibitor concentration ($i$).

## Acknowledgements

This study was supported by a research grant from Nakatomi Foundation.

# Additional information

### Funding

| Funder | Grant reference number | Author |
| --- | --- | --- |
| Nakatomi Foundation | | Tomoya Yasujima |

The funders had no role in study design, data collection and interpretation, or the decision to submit the work for publication.

### Author contributions

Yoshihisa Mimura, Conceptualization, Formal analysis, Investigation, Writing – original draft; Tomoya Yasujima, Conceptualization, Formal analysis, Funding acquisition, Validation, Investigation, Visualization, Methodology, Writing – original draft, Project administration; Katsuhisa Inoue, Conceptualization, Investigation, Writing – original draft, Project administration; Shogo Akino, Chitaka Namba, Yutaro Sekiguchi, Investigation; Hiroyuki Kusuhara, Conceptualization, Writing - review and editing; Kinya Ohta, Investigation, Writing - review and editing; Takahiro Yamashiro, Methodology; Hiroaki Yuasa, Conceptualization, Supervision, Writing - review and editing

### Author ORCIDs

Tomoya Yasujima ⓘ https://orcid.org/0000-0003-3863-5063
Hiroaki Yuasa ⓘ http://orcid.org/0000-0001-8349-6694

Reviewer #1 (Public review): https://doi.org/10.7554/eLife.98853.3.sa1
Reviewer #2 (Public review): https://doi.org/10.7554/eLife.98853.3.sa2
Reviewer #3 (Public review): https://doi.org/10.7554/eLife.98853.3.sa3
Author response https://doi.org/10.7554/eLife.98853.3.sa4

# Additional files

### Supplementary files

• MDAR checklist

### Data availability

All data generated or analysed during this study are included in the manuscript.

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
