## [Editor Report · eLife Assessment]

This work identifies the molecular function of an orphan human transporter, SLC35G1, providing **convincing** evidence that this protein is involved in intestinal citrate absorption. This work provides **important** insight into transporter function and human physiology.

---

## [Referee Report · Reviewer #1 (Public review)]

Summary:

The current manuscript provides solid evidence that the molecular function of SLC35G1, an orphan human SLC transporter, is citrate export at the basolateral membrane of intestinal epithelial cells. Multiple lines of evidence, including radioactive transport experiments, immunohistochemical staining, gene expression analysis, and siRNA knockdown are combined to deduce a model of the physiological role of this transporter.

Strengths:

The experimental approaches are comprehensive, and together establish a strong model for the role of SLC35G1 in citrate uptake. The observation that chloride inhibits uptake suggests an interesting mechanism that exploits the difference in chloride concentration across the basolateral membrane.

Weaknesses:

A gap in this study is that the mechanism of the transporter has not been established. The authors propose that the mechanism is facilitated diffusion, while also leaving open the possibility that citrate transport is coupled to another ion, such as chloride. However, another result from this study seems to be in conflict with the proposed facilitative diffusion mechanism. Specifically, the study finds that uptake is not impacted by membrane depolarization. This would imply that transport is not electrogenic, whereas facilitated diffusion of citrate anion should be an electrogenic process.

---

## [Referee Report · Reviewer #2 (Public review)]

Summary:

The primary goal of this study was to identify the transport pathway that is responsible for the release of dietary citrate from enterocytes into blood across the basolateral membrane.

Strengths:

The transport pathway responsible for the entry of dietary citrate into enterocytes was already known, but the transporter responsible for the second step remained unidentified. The studies presented in this manuscript identify SLC35G1 as the most likely transporter that mediates the release of absorbed citrate from intestinal cells into the serosal side. This fills an important gap in our current knowledge on the transcellular absorption of dietary citrate. The exclusive localization of the transporter in the basolateral membrane of human intestinal cells and the human intestinal cell line Caco-2 and the inhibition of the transporter function by chloride support this conclusion.

Weaknesses:

(i) The substrate specificity experiments have been done with relatively low concentrations of potential competing substrates, considering the relatively low affinity of the transporter for citrate. Given that NaDC1 brings in not only citrate as a divalent anion and also other divalent anions such as succinate, it is possible that SLC35G1 is responsible for the release of not only citrate but also other dicarboxylates. However the substrate specificity studies show that the dicarboxylates tested did not compete with citrate, meaning that SLC35G1 is selective for the citrate (2-), but this conclusion might be flawed because of the low concentration of the competing substrates used in the experiment. Furthermore, the apical NaDC1 is not selective for citrate; in fact, it transports citrate with a much lower affinity than it transports dicarboxylates such as succinate. If what the authors suggest that SLC35G1 is selective for citrate is correct, there must be another transporter for the efflux of dicarboxylates. The authors should have performed a dose-response experiment for the dicarboxylates tested as potential substrates before making the conclusion that SLC35G1 is selective for citrate.

(ii) The authors have used MDCK cells for assessment of the transcellular transfer of citrate via SLC35G1, but it is not clear whether this cell line expresses NaDC1 in the apical membrane as the enterocytes do. Even though the authors expressed SLC35G1 ectopically in MDCK cells and showed that the transporter localizes to the basolateral membrane, the question as to how citrate actually enters the apical membrane for SLC35G1 in the other membrane to work remains unanswered.

(iii) The role of chloride in the efflux of citrate remains not evaluated in detail. Similarly, the potential role of membrane potential in the transport function of SLC35G1 remains unknown. Since the SLC35G1-mediated uptake appears to be similar in the presence and absence of potassium, the authors argue that membrane potential has no role in the transport process. Since it is proposed that the divalent citrate is the substrate for the transporter, it is difficult to reconcile with the conclusion that the membrane potential has no impact on the transport process, especially given the fact that no other exchangeable anion has been shown or suggested. Even if chloride is the potential exchangeable anion, it still begs the question as to the stoichiometry of citrate:chloride if membrane potential plays no role. Obviously, additional work is needed to figure out the actual transport mechanism for SLC35G1.

---

## [Referee Report · Reviewer #3 (Public review)]

The authors convincingly show that SLC35G1 mediates uptake of citrate which is dependent on pH and chloride concentration. Putting their initial findings in a physiological context, they present human tissue expression data of SLC35G. Their Transwell assay indicates that SLC35G1 is a citrate exporter at the basolateral membrane.

Weaknesses:

The manuscript would benefit from the inclusion of the antibody validation results. Related to the localization of SLC35G1, the polyclonal antibody was not validated in the knockdown cells used in the study. This would strengthen the antibody validation, the localization results as well as the transport assay in 2C.

Also, it is unclear why the Transwell assay was not performed upon knockdown of SLC35G1 to support the conclusions.

---

## [Author Response]

The following is the authors’ response to the original reviews.

**Public Reviews:**

**Reviewer #1 (Public Review):**
Summary:The current manuscript provides strong evidence that the molecular function of SLC35G1, an orphan human SLC transporter, is citrate export at the basolateral membrane of intestinal epithelial cells. Multiple lines of evidence, including radioactive transport experiments, immunohistochemical staining, gene expression analysis, and siRNA knockdown are combined to deduce a model of the physiological role of this transporter.Strengths:The experimental approaches are comprehensive, and together establish a strong model for the role of SLC35G1 in citrate uptake. The observation that chloride inhibits uptake suggests an interesting mechanism that exploits the difference in chloride concentration across the basolateral membrane.Weaknesses:Some aspects of the results would benefit from a more thorough discussion of the conclusions and/or model.For example, the authors find that SLC35G1 prefers the dianionic (singly protonated) form of citrate, and rationalize this finding by comparison with the substrate selectivity of the citrate importer NaDC1. However, this comparison has weaknesses when considering the physiological pH for SLC35G1 and NaDC1. NaDC1 binds citrate at a pH of ~5.4 (the pKa of citrate is 5.4, so there is a lot of dianionic citrate present under physiological circumstances). SLC35G1 binds citrate under pH conditions of ~7.5, where a very small amount of dianionic citrate is present. The data clearly show a pH dependence of transport, and the authors rule out proton coupling, but the discrepancy between the pH dependence and the physiological expectations should be addressed/commented on.

Thank you for your insightful comment. Citrate exists mostly in its trianionic form under near neutral pH conditions in biological fluids, as you pointed out. Its dianionic form represents only a small portion (about 1/100) of total citrate due to the pKa. However, significant SLC35G1-specific uptake was observed under near neutral pH conditions (Figure 1G). Therefore, although SLC35G1-mediated citrate transport is less efficient under physiologically relevant near neutral pH conditions, it could still play a role particularly in the intestinal absorption process, in which the concentration gradient of dianionic citrate could be maintained by continuous supply by NaDC1-mediated apical uptake.

The rationale for the series of compounds tested in Figure 1F, which includes metabolites with carboxylate groups, a selection of drugs including anion channel inhibitors and statins, and bile acids, is not described. Moreover, the lessons drawn from this experiment are vague and should be expanded upon. It is not clear what, if anything, the compounds that reduce citrate uptake have in common.

Thank you for highlighting the need for clarity regarding the compounds tested in Figure 1F. The tested compounds were TCA cycle intermediates (fumarate, α-ketoglutarate, malate, pyruvate, and succinate) as substrate candidate carboxylates analogous to citrate, diverse anionic compounds (BSP, DIDS, probenecid, pravastatin, and taurocholate) as those that might be substrates or inhibitors, and diverse cationic compounds (cimetidine, quinidine, and verapamil) as those that are least likely to interact with SLC35G1. Among them, certain anionic compounds significantly reduced SLC35G1-specific citrate uptake, suggesting that they may interact with SLC35G1. However, we could not identify any structural features commonly shared by these compounds, except that they have anionic moieties. We acknowledge that it requires further elaboration to clarify such structural features. We have revised the relevant section on p. 3 (line 25 - 32) to include these.

The transporter is described as a facilitative transporter, but this is not established definitively. For example, another possibility could involve coupling citrate transport to another substrate, possibly even chloride ion.

Thank you for your insightful comment regarding the nature of SLC35G1's transport mechanism. While we have described SLC35G1 as a facilitative transporter based on our current data, we acknowledge that this has not been definitively proven, as you pointed out, and we cannot exclude the possibility that its sensitivity to extracellular Cl- might imply its operation as a citrate/Cl- exchanger. To examine the possibility, we would need to manipulate the chloride ion gradient across the plasma membrane. Particularly, generating an outward Cl- gradient to see if it could enhance citrate uptake could be a potential strategy. However, current techniques do not allow us to effectively generate the Cl- gradient, thus preventing us from conclusively verifying this possibility. We recognize the importance of further investigating this aspect in future studies. Your suggestion highlights an important area for additional research to fully understand the transport mechanism of SLC35G1. We have additionally commented on this issue on p. 4 (line 1 – 3).

**Reviewer #2 (Public Review):**
Summary:The primary goal of this study was to identify the transport pathway that is responsible for the release of dietary citrate from enterocytes into blood across the basolateral membrane.Strengths:The transport pathway responsible for the entry of dietary citrate into enterocytes was already known, but the transporter responsible for the second step remained unidentified. The studies presented in this manuscript identify SLC35G1 as the most likely transporter that mediates the release of absorbed citrate from intestinal cells into the serosal side. This fills an important gap in our current knowledge of the transcellular absorption of dietary citrate. The exclusive localization of the transporter in the basolateral membrane of human intestinal cells and the human intestinal cell line Caco-2 and the inhibition of the transporter function by chloride support this conclusion.Weaknesses:(i) The substrate specificity experiments have been done with relatively low concentrations of potential competing substrates, considering the relatively low affinity of the transporter for citrate. Given that NaDC1 brings in not only citrate as a divalent anion but also other divalent anions such as succinate, it is possible that SLC35G1 is responsible for the release of not only citrate but also other dicarboxylates. But the substrate specificity studies show that the dicarboxylates tested did not compete with citrate, meaning that SLc35G1 is selective for the citrate (2-), but this conclusion might be flawed because of the low concentration of the competing substrates used in the experiment.

Thank you for your valuable comment on our substrate specificity experiments. As you pointed out, we cannot rule out the possibility that dicarboxylates might be recognized by SLC35G1 with low affinity as the tested concentration was relatively low. However, at the concentration of 200 μM, competing substrates with an affinity comparable to that of citrate could inhibit SLC35G1-specific citrate uptake by about 30%. Therefore, it is likely that the compounds that did not exhibit significant effect have no affinity or at least lower affinity than citrate to SLC35G1. Further studies should explore a broader range of concentrations for potential substrates including those with lower affinity. It would help clarify the substrate recognition characteristics of SLC35G1 and if it indeed has a unique preference for citrate over dicarboxylates. We have additionally mentioned that on p. 3, line 32 – 35.

(ii) The authors have used MDCK cells for assessment of the transcellular transfer of citrate via SLC35G1, but it is not clear whether this cell line expresses NaDC1 in the apical membrane as the enterocytes do. Even though the authors expressed SLC35G1 ectopically in MDCK cells and showed that the transporter localizes to the basolateral membrane, the question as to how citrate actually enters the apical membrane for SLC35G1 in the other membrane to work remains unanswered.

Thank you for highlighting this important aspect of our study. The mechanism of apical citrate entry in MDCKII cells is unknown, although NaDC1 or a similar transporter may be involved. However, this set of experiments have successfully demonstrated the basolateral localization of SLC35G1 and its operation for citrate efflux. Attempts to clarify the apical entry mechanism may need to be included in future studies for more detailed characterization of the model system using MDCKII cells. This would help in fully understanding the transcellular transport system for citrate. Investigation using Caco-2 cells or MDCKII cells double transfected with NaDC1 and SLC35G1 would also need to be induced in future studies to gain more definitive insights into the transcellular transport mechanism for citrate in the intestine, delineating the suggested cooperative role of NaDC1 and SLC35G1. We would be grateful for your understanding of our handling regarding this issue.

(iii) There is one other transporter that has already been identified for the efflux of citrate in some cell types in the literature (SLC62A1, PLoS Genetics; 10.1371/journal.pgen.1008884), but no mention of this transporter has been made in the current manuscript.

Thank you for bringing up the relevance of SLC62A1, which has recently been identified as a citrate efflux transporter in some cell types (PLoS Genet, 16, e1008884, 2020). We have now included comments on this transporter in Introduction (p. 2).

**Reviewer #3 (Public Review):**
Summary:Mimura et al describe the discovery of the orphan transporter SLC35G1 as a citrate transporter in the small intestine. Using a combination of cellular transport assays, they show that SLC35G1 can mediate citrate transport in small intestinal cell lines. Furthermore, they investigate its expression and localization in both human tissue and cell lines. Limited evidence exists to date on both SLC35G1 and citrate uptake in the small intestine, therefore this study is an important contribution to both fields. However, the main claims by the authors are only partially supported by experimental evidence.Strengths:The authors convincingly show that SLC35G1 mediates uptake of citrate which is dependent on pH and chloride concentration. Putting their initial findings in a physiological context, they present human tissue expression data of SLC35G. Their Transwell assay indicates that SLC35G1 is a citrate exporter at the basolateral membrane.Weaknesses:Further confirmation and clarification are required to claim that the SLC indeed exports citrate at the basolateral membrane as concluded by the authors. Most experiments measure citrate uptake, but the authors state that SLC35G1 is an exporter, mostly based on the lack of uptake at physiological conditions faced at the basolateral side. The Transwell assay in Figure 1L is the only evidence that it indeed is an exporter. However, in this experiment, the applied chloride concentration was not according to the proposed model (120 mM at the basolateral side). The Transwell assay, or a similar assay measuring export instead of import, should be carried out in knockdown cells to prove that the export indeed occurs through SLC35G1 and not through an indirect effect. Related to the mentioned chloride sensitivity, it is unclear how the proposed model works if the SLC faces high chloride conditions under physiological conditions though it is inhibited by chloride.

Thank you for highlighting these important points. We used the Cl--rich medium in transcellular transport studies, as stated in the relevant section in Meterials and Methods (p. 6, line 2 – 5). The Cl- concentration (144 mM) was comparable to the physiological concentration in extracellular body fluids. To clarify that experimental condition, we have additionally noted that in the text (p. 4, line 9) and the legends of Figs. 1K and 1L. The results indicate that basolaterally localized SLC35G1 can mediate citrate export effectively under the Cl--rich extracellular condition. The transport mechanism regulated by Cl- is unclear, but it is difficult to further clarify the mechanism at this time. We recognize the importance of further investigating the aspect in future studies, including the possibility that SLC35G1 might be a citrate/Cl- exchanger, as pointed out by Reviewer #1 (3rd comment).

**Recommendations for the authors:**

**Reviewer #1 (Recommendations For The Authors):**
The figures are very tiny and difficult to see. The inset in Figure 1C is much too small to be readable. I suggest enlarging the panels.

Thank you for your feedback. As advised, we have enlarged the panels to improve visibility.

Line 74: "certain anionic compounds signficantly inhibited SLC35G1-specific citrate uptake, indicating they are also recognized by SLC35G1." This sentence should be reworded since the mechanism is not clear. The word "reduced" would be a better option than "inhibited." Are there other interpretations besides SLC35G1 binding to explain the observations?

Thank you for your suggestion. We have reworded the sentence to improve clarity (p. 3, line 30). It may be possible to speculate that they interact with SLC35G1, but the mechanisms are not clear yet.

The manuscript is vague about how the transporter was discovered. If a screen of orphan transporters was performed to identify a citrate transporter, this should be described.

Thank you for pointing out the need for more details regarding the discovery of the transporter. We have added some detailed description at the beginning of Results and Discussion (p. 3).

**Reviewer #2 (Recommendations For The Authors):**
Recommendations for the authors:(1) For transcellular transport of citrate and the role of SLC35G1, it would be better to use Caco-2 cells cultured on Transwells because these cells express NaDC1 in the apical membrane and the authors have shown that SLC35G1 is expressed in the basolateral membrane in this cell line. The mechanism for the entry of citrate into MDCK cells used in the present manuscript is not known. If the authors prefer to use MDCK cells because of their superior use for polarization, they can use a double transfection (NaDC1 and SLC35G1) to differentially express the two transporters in the apical versus and basolateral membrane and then use the cells for trans cellular transport of citrate.

Please refer to our reply to your second review comment.

(2) The substrate specificity experiments should use concentrations higher than 0.2 mM for competing dicarboxylates because the Km for citrate is only 0.5 mM. It is likely that NaDC1 brings in citrate and other dicarboxylates into enterocytes and then SLC35G1 mediates the efflux of these metabolic intermediates into blood.

Please refer to our reply to your first review comment.

(3) One major aspect of the transport function of this newly discovered citrate efflux transporter that has not been explored is the role of membrane potential in the transport function. The transporter is not coupled to Na or K or even H; so then the transport of citrate via this transporter must be electrogenic. Of course, this would be perfect for the transporter to function in the efflux of citrate because of the inside-negative membrane potential, but the authors need to show that the transporter is electrogenic. This can be examined through Caco-2 cells and/or MDCK cells expressing SLC35G1 and examining the impact of changes in membrane potential (valinomycin and K) on the transport of citrate.

Thank you for your suggestion. As shown in Figure 1D, the use of K-gluconate in place of Na-gluconate, which induces plasma membrane depolarization, had no impact on the specific uptake of citrate, suggesting that SLC35G1-mediated citrate transport is independent of membrane potential. We have additionally mentioned this on p. 3 (line 21 – 24).

(4) The localization studies mention Na/K ATPase component as a basolateral membrane marker, but the text describes it as BCRP. This needs to be corrected.

Thank you for pointing out the mistake. We have corrected that. The marker was ATP1A1.

**Reviewer #3 (Recommendations For The Authors):**
Major points:(1) Most experiments measure citrate uptake, but the authors state that SLC35G1 is an exporter, mostly based on the lack of uptake at physiological conditions faced at the basolateral side. The Transwell assay in Figure 1L is the only evidence that it indeed is an exporter. However, in this experiment, the applied chloride concentration was not according to the proposed model (120mM at basolateral side). Why was this chloride concentration not mimicked accordingly in the Transwell assay?(2) The Transwell assay, or a similar assay measuring export instead of import, should be carried out in knockdown cells to prove that the export indeed occurs through SLC35G1 and not through an indirect effect.(3) Related to the mentioned chloride sensitivity, it is unclear how the proposed model works if the SLC faces high chloride conditions under physiological conditions though it is inhibited by chloride.

Please refer to our reply to your review comments.

Related to the localization of SLC35G1:(4) The polyclonal antibody against SLC35G1 should be validated to prove the specificity. This should be relatively straightforward given the authors have SLC35G1 knockdown cells.

Thank you for your suggestion. To validate the specificity of the polyclonal antibody against SLC35G1, we prepared HEK293 cells transiently expressing SLC35G1 and SLC35G1 tagged with a FLAG epitope at the C-terminus (SLC35G1-FLAG). In the immunostained images, whereas only SLC35G1-FLAG was stained with the anti-FLAG antibody, both SLC35G1 and SLC35G1-FLAG were stained with the anti-SLC35G1 antibody, indicating that the anti-SLC35G1 antibody can recognize SLC35G1. In addition, the localization patterns of SLC35G1-FLAG observed with both antibodies were consistent, indicating furthermore that the anti-SLC35G1 antibody can recognize SLC35G1 specifically. Based on all these, the specificity of the anti-SLC35G1 antibody was validated.

**Author response image 1. sa4fig1:** 

*(5) To strengthen the data on the localization of SLC35G1, the cell lines should be co-stained with a plasma membrane marker as well, not just in tissue with ATP1A1. In polarized cells co-staining with apical and basolateral markers should be applied.*

SLC35G1 was indicated to be localized to the basolateral membrane geometrically in both polarized MDCKII and Caco-2 cells. This finding aligns with its basolateral localization indicated by its colocalization with ATP1A1 in the human small intestinal section. These results are we consider sufficient to support the basolateral localization characteristics of SLC35G1.

General points:(6) In the abstract the authors mention that they focus on highly expressed orphan transporters in the small intestine as candidates. However, no other candidates are mentioned or discussed in the study. Consequently, this should be rephrased.

Thank you for the advice. Also taking into consideration the third recommendation point by Reviewer #1, we have added some detailed description at the beginning of Results and Discussion (p. 3).

(7) As far as mentioned there is exactly one (other) publication on SLC35G1 (10.1073/pnas.1117231108). The authors should discuss this only publication with functional data on SLC35G1 in more detail. How do the authors integrate their findings with the existing knowledge? For example, why did the authors not investigate the impact of Ca2+ on SLC35G1 transport?

Thank you for your suggestion. SLC35G1 was indicated to be mainly localized to the endoplasmic reticulum (ER) in the earlier study, in which SLC35G1 was tagged with GFP. A possibility is that SLC35G1 was wrongly directed to ER due to the modulation in the study. We have additionally mentioned this possibility in the relevant section (p. 3, line 9 – 11). We have also revised a relevant sentence on p. 3 (line 5).

With regard to another point that GFP-tagged SLC35G1 was indicated to interact with STIM1, we examined its effect on SLC35G1-mediated citrate uptake supplementary. As shown in the accompanying figure, coexpression of HA-tagged STIM1 did not affect the elevated citrate uptake induced by FLAG-tagged SLC35G1, indicating that STIM1 has no impact on citrate transport function of SLC35G1 at the plasma membrane.

**Author response image 2. sa4fig2:** (A) Effect of the coexpression of HA-tagged STIM1 on [14C]citrate (1 μM) uptake by FLAG-tagged SLC35G1 transiently expressed in HEK293 cells. The uptake was evaluated for 10 min at pH 5.5 and 37°C. Data represent the mean ± SD of three biological replicates. Statistical differences were assessed using ANOVA followed by Dunnett’s test. *, *p* < 0.05 compared with the control (gray bar). (B) Western blot analysis was conducted by probing for the HA and FLAG tags, using the whole-cell lysate samples (10 µg protein aliquots) prepared from cells expressing HA-STIM1 and/or FLAG-SLC35G1. The blots of β-actin are shown for reference.

(8) Generally, the introduction could provide more background.

In response to your suggestion and also to the third review comment from Reviewer #2, we have now additionally included comments on SLC62A1, which has recently been reported as a citrate efflux transporter in some cell types, in Introduction.

Minor points:(9) There is a typo in Figure 1D: manniotol instead of mannitol.

Thank you for pointing that out. We have corrected the typo in Figure 1D.

(10) Figure 1J: The resolution is low and the localization to the basolateral membrane is not conclusive based on this image. It seems rather localized at the whole membrane and intracellularly too.

Thank you for your feedback. We have enhanced the resolution of the image and also enlarged it to improve clarity and make the basolateral membrane localization more discernible.

(11) Figure 1K: Clarification is needed if the experiment was performed in the Transwell plate. Based on the results from the pH titration experiment, it is expected that there is no uptake at pH7.4. Therefore, this experiment does not seem to provide additional evidence or support the conclusions drawn related to cellular polarization.

Please refer to our reply to your review comments.